# Efficient Gaussian process-based motor hotspot hunting with concurrent optimization of TMS coil location and orientation

David Luis Schultheiss[1,2], Zsolt Turi[3], Joschka Boedecker[1,2‡], Andreas Vlachos[2,3,4‡*]

**1** Neurobotics Lab, Department of Computer Science, University of Freiburg, Freiburg, Germany, **2** Center BrainLinks-BrainTools, University of Freiburg, Freiburg, Germany, **3** Department of Neuroanatomy, Institute of Anatomy and Cell Biology, University of Freiburg, Freiburg, Germany, **4** Center for Basics in NeuroModulation (NeuroModulBasics), University of Freiburg, Freiburg, Germany

‡ Co-last authorship.
* andreas.vlachos@anat.uni-freiburg.de

## Abstract

Transcranial magnetic stimulation (TMS) is a widely used non-invasive brain stimulation technique in neuroscience research and clinical applications. TMS-based motor hotspot hunting describes the process of identifying the optimal scalp location to elicit robust and reliable motor responses. It is critical to ensure reproducibility of TMS parameters, as well as to determine safe and precise stimulation intensities in both healthy participants and patients. Typically, this process targets motor responses in contralateral short hand muscles. However, hotspot hunting remains challenging due to the vast parameter space and time constraints. To address this, we present an approach that concurrently optimizes both spatial and angular TMS parameters for hotspot hunting using Gaussian processes and Bayesian optimization. We systematically evaluated five state-of-the-art acquisition functions on electromyographic TMS data from eight healthy individuals enhanced by simulated data from generative models. Our results consistently demonstrate that optimizing spatial and angular TMS parameters simultaneously enhances the efficacy and spatial precision of hotspot hunting. Furthermore, we provide mechanistic insights into the acquisition function behavior and the impact of coil rotation constraints, revealing critical limitations in current hotspot-hunting strategies. Specifically, we show that arbitrary constraints on coil rotation angle are suboptimal, as they reduce flexibility and fail to account for individual variability. We further demonstrate that acquisition functions differ in sampling strategies and performance. Functions overly emphasizing exploitation tend to converge prematurely to local optima, whereas those balancing exploration and exploitation—particularly Thompson sampling—achieve superior performance. These findings highlight the importance of acquisition function selection and the necessity of removing restrictive coil rotation constraints for effective hotspot hunting. Our work advances TMS-based hotspot identification, potentially reducing

**Data availability statement:** The code for this paper will be publicly available at https://github.com/david-schu/gp-tms-hsh. The electromyographic data used in this study originates from a secondary analysis of the original data collected in our previous study https://doi.org/10.1101/2025.02.20.639076. No new data was collected from human participants for this study.

**Funding:** This work is part of BrainLinks-BrainTools which is funded by the Federal Ministry of Economics, Science and Arts of Baden-Württemberg within the sustainability program for projects of the excellence initiative II. The study was supported by Federal Ministry of Education and Research, Germany (BMBF, 01GQ2205A to AV). The funders had no role in study design, data collection and analysis, decision to publish, or manuscript preparation.

**Competing interests:** The authors have declared that no competing interests exist.

participant burden and improving safety in both research and clinical applications beyond the motor cortex.

## Author summary

Transcranial magnetic stimulation (TMS) is a non-invasive technique used to stimulate the human brain. It is widely applied in both neuroscience research and clinical practice. A key step in many TMS procedures is hotspot hunting, which involves finding the optimal spot on the scalp to produce muscle responses. This process is often slow and relies on manual adjustments of the stimulation settings. In this study, we developed a method that automatically optimizes the position and rotation of the TMS coil using advanced techniques such as Gaussian processes and Bayesian optimization. We tested our approach on data from healthy participants and supported it with realistic computer simulations. Our results show that our method improves both the speed and accuracy of hotspot detection. We also found that certain choices in the optimization process—especially how uncertainty is managed—have a strong effect on performance. In particular, we show that common restrictions on coil rotation can reduce effectiveness and should be reconsidered. This work provides a step toward making TMS faster, safer, and more reliable for both research and clinical use.

## 1 Introduction

Transcranial magnetic stimulation (TMS) is a non-invasive brain stimulation technique that generates transient intracranial electric fields (E-field) via electromagnetic induction, enabling external modulation of neuronal activity and brain functions [1–8]. Single-pulse and repetitive TMS (rTMS) are widely employed in basic neuroscience research to induce and investigate neuromodulatory effects across multiple biophysical scales, including molecular, cellular, microcircuit, network, and behavioral levels [9–21]. Clinically, TMS is used for preoperative functional brain mapping and as a therapeutic intervention for neurological and psychiatric disorders [22–34].

TMS-based motor hotspot hunting (hereafter referred to as hotspot hunting) is a critical procedure for identifying the optimal scalp location that elicits the most robust and reliable motor responses, typically recorded from the contralateral short hand muscles [35]. Traditionally, this function-guided mapping has been performed manually by trained operators without neuronavigation [36,37]. However, manual hotspot hunting relies on implicit operator strategies for coil positioning, leading to high inter-operator variability and limited reproducibility.

A more standardized approach is grid search, where predefined equidistant scalp locations (e.g., Cartesian grid) with fixed coil rotation angles are systematically tested, with or without neuronavigation [24,36,38–56]. More advanced methods have been developed, including individual sulcus aligned neuronavigation, mapping algorithms (e.g., pseudo-random walk), closed-loop optimization, and robotic assistance [57–62]. However, these methods share key limitations, such as exhaustive and time-consuming searches, manual coil adjustments, and restricted coil rotation

angles [63]. Such constraints could lead to suboptimal hotspot hunting, increasing the number of trials required, reducing spatial precision, and often necessitating higher stimulation intensities. Consequently, this increases participant burden, prolongs hotspot hunting, and reduces overall comfort.

Gaussian processes (GPs) have recently emerged as powerful tools for modeling complex nonlinear systems with limited data [64]. Combined with Bayesian optimization, GPs act as surrogate models that balance exploration and exploitation, enabling efficient sampling in high-dimensional parameter spaces [65,66]. This makes them particularly well-suited for closed-loop applications such as hotspot hunting, where exhaustive search is impractical and sample efficiency is crucial [67]. Additionally, GPs naturally incorporate prior knowledge (e.g., previous motor-evoked potentials; MEPs) and provide uncertainty estimates, making them robust to inter-individual variability [68].

Previous studies have demonstrated that GPs can substantially improve hotspot hunting by optimizing stimulation parameters [69–72]. However, these implementations remain limited—for example certain parameters such as coil rotation angles are often fixed arbitrarily (e.g., 45°) [71,72]. While some approaches have attempted to optimize spatial and angular parameters separately, they still fail to explore the full candidate space, limiting search efficiency [70]. Additionally, prior Bayesian neuronavigation methods have often constrained coil location to discrete grids or predefined patterns and relied on acquisition functions that may be suboptimal, limiting both sampling efficiency and mechanistic insight.

In this study, we introduce a novel closed-loop hotspot hunting procedure that simultaneously optimizes *both* spatial and angular TMS parameters using GPs and Bayesian optimization. Unlike previous methods [59,69,70], which restrict coil orientation, confine the search to discrete locations, or tune parameters separately, our approach jointly explores position and orientation in a continuous parameter space. It further leverages a broader range of acquisition functions beyond entropy-based methods, resulting in more efficient sampling and faster convergence. To our knowledge, this is the first approach to jointly optimize these parameters, overcoming key limitations observed in previous studies. We rigorously evaluated our method on a TMS MEP dataset from eight healthy individuals [73]. Our results demonstrate that this approach significantly enhances efficacy, spatial precision, and reproducibility in hotspot hunting. By improving TMS targeting, this method has the potential to advance both research and clinical applications, including motor mapping, presurgical functional planning, and therapeutic interventions.

## 2 Methods

### 2.1 Gaussian processes

The goal is to find the stimulation parameters that produce the largest reliable motor response. We model this relationship of stimulation parameters $\mathbf{s}$ and response $r$ for as

$$r = f(\mathbf{s}) + \epsilon, \tag{1}$$

where $f$ is a unknown response function, and $\epsilon \sim \mathcal{N}(0, \sigma_\epsilon^2)$ is noise, representing the variability of MEP amplitude measurements. Because we assume $\epsilon$ to be statistically independent of $\mathbf{s}$, the problem of finding the most responsive parameter settings reduces to:

$$\hat{\mathbf{s}} = \arg\max_{\mathbf{s}} f(\mathbf{s}). \tag{2}$$

In previous studies [70,72], Gaussian processes [64] have proven to be a suitable choice for modeling the response function $f$. A GP is a collection of random variables, any finite number of which have a joint Gaussian distribution. It defines a distribution over functions:

$$f(\mathbf{s}) \sim \mathcal{GP}(\mu(\mathbf{s}), k(\mathbf{s}, \mathbf{s}')), \tag{3}$$

where $(\mathbf{s}, \mathbf{s}')$ are input points, $\mu(\mathbf{s})$ is the prior mean function, $k(\mathbf{s}, \mathbf{s}')$ is the *covariance function* or *kernel*, which defines the covariance between function values at points $\mathbf{s}$ and $\mathbf{s}'$.

For a set of training points $\mathbf{S} = \{\mathbf{s}_1, \mathbf{s}_2, \ldots, \mathbf{s}_n\}$ and their corresponding response values $\mathbf{r} = \{r_1, r_2, \ldots, r_n\}$, the joint distribution is multivariate Gaussian:

$$\mathbf{f} \sim \mathcal{N}(\mathbf{0}, K(\mathbf{S}, \mathbf{S})) \tag{4}$$

where $K(\mathbf{S}, \mathbf{S})$ is the covariance matrix with entries $K_{ij} = k(\mathbf{s}_i, \mathbf{s}_j)$.

The predictive distribution of the function value $f(\mathbf{s})$ at a new test point $\mathbf{s}$ is also Gaussian:

$$f(\mathbf{s})|\mathbf{S}, \mathbf{r} \sim \mathcal{N}(\mu_r(\mathbf{s}), \sigma_r(\mathbf{s})) \tag{5}$$

Typically the GP prior has zero mean, hence the posterior mean $\mu_r$ and variance $\sigma_r$ are given as:

$$\mu_r(\mathbf{s}) = k(\mathbf{s}, \mathbf{S})\left(K(\mathbf{S}, \mathbf{S}) + \sigma_\epsilon^2 \mathbf{I}\right)^{-1} \mathbf{r} \tag{6}$$

$$\sigma_r(\mathbf{s}) = k(\mathbf{s}, \mathbf{s}) - k(\mathbf{s}, \mathbf{S})\left(K(\mathbf{S}, \mathbf{S}) + \sigma_\epsilon^2 \mathbf{I}\right)^{-1} k(\mathbf{S}, \mathbf{s}) \tag{7}$$

Because the measured responses are strictly non-negative, the assumption of a Gaussian distribution of the posterior is not a reasonable modeling assumption here. Thus, we fit a warped Gaussian process [74] and apply the square-root warping function as proposed by [72] to ensure non-negativity.

## 2.2 Bayesian optimization

Bayesian optimization (BO) uses a probabilistic model to represent an unknown function that is to be optimized. This model is updated iteratively as new data points (function evaluations) are gathered. In the present study, we use a GP as a surrogate model to approximate motor responses as a function of TMS-coil parameters. The combination of stimulation location and orientation makes the input space vast. However, minimizing the number of samples for hotspot hunting is crucial to maximize the samples available for follow-up experiments. Consequently, naive sampling techniques such as grid, spiral, or random sampling are too inefficient and yield suboptimal results. Hence, we investigated more sophisticated acquisition functions.

**Acquisition functions.** In this work, we tested several well-known and state-of-the-art acquisition functions for BO: Upper Confidence Bound (UCB) [75], Expected Improvement (EI) [76,77], Thompson sampling (TS) [78], Knowledge Gradient (KG) [79], and Max-Value Entropy Search (MVE) [80]. All of these acquisition functions aim to balance exploration of the input space and exploitation of highly responsive areas.

**Upper confidence bound.** UCB constructs an optimistic estimate of the unknown function using the Gaussian process posterior. It selects the next point by maximizing:

$$\text{UCB}(\boldsymbol{s}) = \mu_{r,n}(\boldsymbol{s}) + \beta_n^{1/2}\sigma_{r,n}(\boldsymbol{s}). \tag{8}$$

Here, $\mu_{r,n}(\boldsymbol{s})$ and $\sigma_{r,n}(\boldsymbol{s})$ are the posterior mean and standard deviation at step $n$. The exploration-exploitation trade-off is controlled by $\beta_n$, with $\beta_n \sim \exp(-n)$, ensuring sufficient exploration in early sampling stages by sublinear decline of the uncertainty term.

**Expected improvement.** EI measures the expected gain over the current best observation $r^\star$. Given the GP posterior, it is defined as:

$$\text{EI}(\boldsymbol{s}) = \mathbb{E}\left[\max(f(\boldsymbol{s}) - r^\star, 0)\right] \tag{9}$$

For a normally distributed $f(\boldsymbol{s})$, EI has a closed-form expression:

$$\text{EI}(\boldsymbol{s}) = \left(\mu_{r,n}(\boldsymbol{s}) - r^\star\right)\Phi\left(\frac{\mu_{r,n}(\boldsymbol{s}) - r^\star}{\sigma_{r,n}(\boldsymbol{s})}\right) + \sigma_{r,n}(\boldsymbol{s})\varphi\left(\frac{\mu_{r,n}(\boldsymbol{s}) - r^\star}{\sigma_{r,n}(\boldsymbol{s})}\right). \tag{10}$$

Here, $\Phi$ and $\varphi$ are the CDF and PDF of the standard normal distribution.

**Thompson sampling.** TS iteratively selects inputs by sampling a possible realization of the GP posterior:

$$\text{TS}(\boldsymbol{s}) \sim \mathcal{N}(\mu_{r,n}(\mathbf{s}), \sigma_{r,n}(\mathbf{s})), \tag{11}$$

and selecting the input that maximizes this sampled function.

**Knowledge gradient.** The KG acquisition function selects the point $\boldsymbol{s}$ that maximizes the expected improvement in the estimate of the global optimum. At iteration $n$, with the current best function value $r^\star$, KG measures the expected increase in the GP posterior maximum after evaluating $f$ at $s$:

$$\text{KG}(\boldsymbol{s}) = \mathbb{E}\left[\max_{\boldsymbol{s}'}\mu_{r,n+1}(\boldsymbol{s}') - \max_{\boldsymbol{s}'}\mu_{r,n}(\boldsymbol{s}')|\boldsymbol{s}_{n+1} = \boldsymbol{s}\right]. \tag{12}$$

Since exact computation is expensive, approximations are used. KG prioritizes sampling where new information is most valuable for refining the estimated optimum.

**Max-value entropy search.** MVE selects the point $\boldsymbol{s}_n$ that maximizes the expected reduction in uncertainty about the global maximum $f^* = \max_{\boldsymbol{s}} f(\boldsymbol{s})$, by reducing the entropy $H(f^*)$ of the distribution over $f^*$. The MVE acquisition function is:

$$\text{MES}(\boldsymbol{s}) = H(f^*) - \mathbb{E}_{f \sim p(r_n|\boldsymbol{s})}\left[H(f^*|r_n, \boldsymbol{s})\right], \tag{13}$$

Here, $H(f^*|r_n, \boldsymbol{s})$ is the entropy of $f^*$ conditioned on observing $r_n$ at $\boldsymbol{s}$, and $p(r_n|\boldsymbol{s})$ is the GP predictive distribution. Since the posterior of $f^*$ is intractable, MES approximates it using samples from the GP posterior at selected candidate points. The entropy terms are estimated via Monte Carlo methods, and Gaussian predictive distributions make sampling straightforward. The main computational challenge is updating the GP posterior for hypothetical observations, which can be mitigated using approximations or batch evaluations.

## 2.3 Algorithm for concurrent optimization of location and orientation

For this work, we model measured MEP amplitudes as the GP response $r = \text{MEP}$ and the stimulation parameters as input $\boldsymbol{s}$. Specifically, the stimulation parameters consist of coil location $x, y$ and coil rotation angle $\theta$, such that $\boldsymbol{s} = \{x, y, \theta\}$.

**Algorithm 1. Optimization algorithm.**

**input** : $S, AF, N_s$
**output:** $\hat{f}$

```
1  Initialize fn with {10 k-means points; 1 random point};
2  n = 1;
3  while n < Ns do
4  |    select new sample sn = arg maxs AF(s)|fn;
5  |    measure rn at sn;
6  |    add rn to r,  sn to S;
7  |    fit fn+1(s)|S,r;
8  |    n = n + 1;
9  end
```

Thus, our GP models the response as

$$r = f(x, y, \theta) + \epsilon = f(\mathbf{s}) + \epsilon. \tag{14}$$

As kernel function we use the squared exponential kernel:

$$k_{\text{RBF}}(\mathbf{s}, \mathbf{s}') = a \cdot \exp\left(-\frac{\|\mathbf{s} - \mathbf{s}'\|^2}{2l^2}\right). \tag{15}$$

Here, $a$ is the amplitude or overall variance of the kernel, that determines how much $f$ can vary from its mean. The second parameter $l$ is the length scale of the kernel and regularizes the smoothness of $f$. These parameters are optimized with maximum likelihood fitting after every sample.

At iteration $n$, a new set of stimulation parameters $\mathbf{s}$ is chosen with the respective acquisition function $AF$

$$\mathbf{s}_n = \arg\max_{\mathbf{s}} AF(\mathbf{s})|f_n$$
$$AF(\mathbf{s}) \in \{\text{UCB}(\mathbf{s}), \text{EI}(\mathbf{s}), \text{TS}(\mathbf{s}), \text{KG}(\mathbf{s}), \text{MVE}(\mathbf{s})\} \tag{16}$$

where $\hat{f}_n$ is the GP fitted at iteration $n$. The newly retrieved response $r_n$ and the stimulation parameters $\mathbf{s}_n$ are added to the training sets $S$ and $r$ and a new GP $\hat{f}_{n+1}$ is fitted. This procedure is repeated upon termination either by reaching a pre-set amount of stimuli, or an early stopping criterion, e.g. stagnation of model parameters. Importantly, in the optimization procedure $\mathbf{s}$ is optimized freely without any constraints. Thus, the location and orientation of the coil are optimized simultaneously in a continuous parameter space.

## 2.4 Dataset and MEP simulation

In a previous study, TMS trials were recorded from 8 participants' right first dorsal interosseous (FDI) muscle containing 300 coil configuration - MEP amplitude pairs [73]. For each participant, we fit a GP with all 300 samples (hereafter referred to as the ground truth) and define the maximum of its posterior as the estimate of the optimal stimulation target. Additionally, for every possible parameter setting $\mathbf{s}$—even if not present in the original dataset—a simulated stochastic response is generated by sampling from the GP posterior predictive distribution defined by the mean $\mu_r(\mathbf{s})$ and variance $\sigma_r(\mathbf{s})$ of the ground truth model. This probabilistic generative model captures both the expected MEP amplitude and its uncertainty at arbitrary stimulation settings, allowing realistic simulation of response variability informed by the empirical data. GP regression has been rigorously applied in the context of TMS-MEP modeling and motor cortex mapping in several prior studies [70–72]. Due to practical constraints preventing new data collection, we rely on these established approaches and their demonstrated validity to underpin our simulation pipeline.

## 2.5 Evaluation metrics

The goal is to compare the ground truth GP predictive function $f_{GT}$ and a fitted GP predictive function $\hat{f}$, retrieved from subsampling the complete parameter space. To quantify the performance of the acquisition functions, we calculate test metrics for a given set of test points $\boldsymbol{S}$. The test points consist of a circular grid of stimulation locations spaced at 3mm distance and centered at the MNI-coordinate of each subjects muscle representation. For each location, we tested rotation angles from 0° to 180° at 20° intervals. This results in a total of $\sim 2,800$ test points. The overall fit of the GP is calculated as the normalized root mean square error (NRMSE):

$$\text{NRMSE}(f_{GT}, \hat{f}) = \sqrt{\frac{1}{N}\sum_{i=1}^{N}\left(\frac{\mu_{GT}(\boldsymbol{s}_i) - \hat{\mu}(\boldsymbol{s}_i)}{\mu_{GT}(\boldsymbol{s}_i)}\right)^2}, \ \boldsymbol{s}_i \in \boldsymbol{S} \tag{17}$$

where $\hat{\mu}(\boldsymbol{s}_i)$ is the posterior mean of the fitted GP given coil parameters $\boldsymbol{s}_i$, and $\mu_{GT}(\boldsymbol{s}_i)$ is the posterior mean of the ground truth GP.

Furthermore, we measure the Euclidean distance between the center of gravity (CoG) of $\hat{\mu}_i$ and $\mu_{i,GT}$. Because the location of the coil and its rotation represent physically different attributes, we calculate the distances for both separately: a location distance $d_{x,y}(\mu_{GT}, \hat{\mu})$, and a rotation distance $d_\theta(\mu_{GT}, \hat{\mu})$. We also only included the top 10 percent of test responses, to diminish the impact of low response areas.

## 3 Results

First, we examined whether concurrently optimizing both spatial and angular TMS parameters improves hotspot hunting. Next, we investigated the impact of k-means initialization on model efficacy across different acquisition functions.Finally, we evaluated the efficacy of the proposed acquisition functions in terms of overall performance and sampling strategies.

**Arbitrary constraints on coil rotation angle are suboptimal for precise hotspot hunting.** Hotspot hunting is commonly performed with a fixed coil rotation angle of 45°. However, this configuration does not necessarily elicit the maximum MEP strength. To quantify the impact of this constraint, we compared the maximum expected return across all coil angles ($r^*_{\text{all}}$) to the maximum expected return at a fixed 45°($r^*_{45°}$). For each participant, we fitted a model $f$ using all 300 stimulation trials $\{\boldsymbol{s}_i, r_i\}$. The maxima were defined as

$$\begin{aligned} r^*_{\text{all}} &= \max_{\boldsymbol{s}_i} \mu(\boldsymbol{s}_i) \\ r^*_{45°} &= \max_{\boldsymbol{s}_i \text{ s.t. } \theta_i = 45°} \mu(\boldsymbol{s}_i), \end{aligned} \tag{18}$$

where $\mu$ denotes the posterior mean of $f$. Across all participants, the median ratio $r^*_{\text{all}}/r^*_{45°}$ was $1.29 \pm 0.53$ (median $\pm$ standard deviation), indicating a 29% increase in the maximum expected return when optimizing both spatial and angular parameters. Individual ratios ranged from 1.10 to 1.47, corresponding a 10% to 47% increase in response strength. Notably, one participant was excluded as an outlier (ratio: 2.83).

**Initialization influences acquisition functions differently.** The performance of acquisition functions is often contingent on the initialization of an informative prior. To assess this effect, we conducted two sets of simulations: one without initialization and one with initialization using 10 k-means clustering points as informative prior. Fig 1 illustrates how initialization impacted the performance of different acquisition functions in Bayesian optimization. EI and UCB exhibited a strong dependence on initialization, as demonstrated by a rapid decrease in NRMSE when an informative prior was provided. Without initialization, their performance remained comparable to random sampling in early iterations and even deteriorated as more samples were acquired. This finding suggested that EI and UCB require structured initial datasets to balance exploration and exploitation effectively. The KG and MVE functions showed a more gradual but consistent

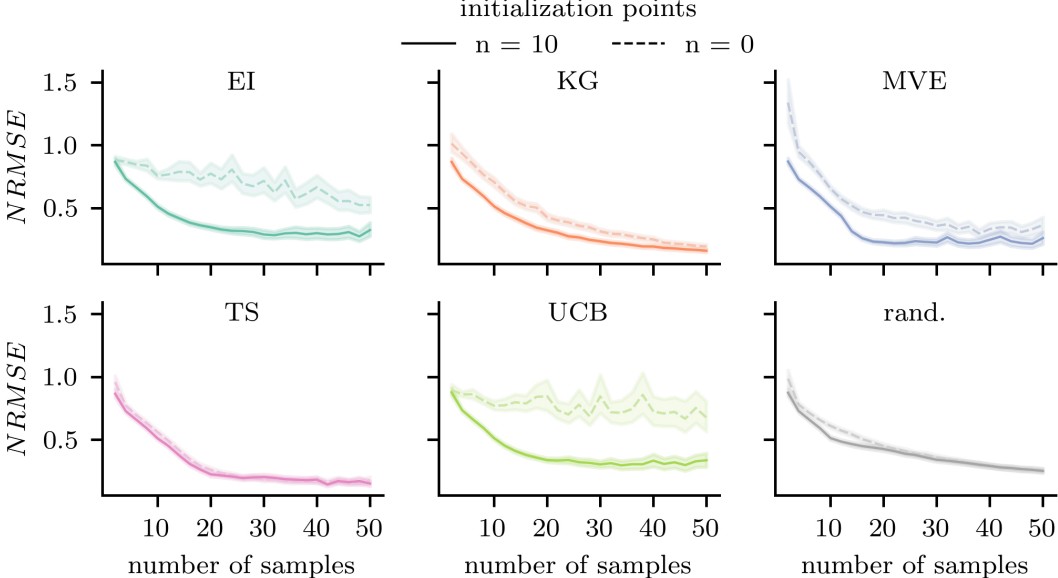

**Fig 1. Effect of initialization on acquisition function performance.** For all eight participants, we subsampled from a total of 300 measured TMS trials and evaluated acquisition function performance under two conditions: without initialization and with initialization using 10 k-means points. Normalized root mean square error (NRMSE) between the ground truth function $f_{GT}$ and the estimated function $\hat{f}$ was measured as the number of stimuli increased. Solid lines represent the mean across 100 random seeds, and shaded areas indicate twice the standard error of the mean. With the exception of Thompson sampling (TS), all acquisition functions exhibited improved performance with k-means initialization (Expected Improvement (EI), Knowledge Gradient (KG), Max-Value Entropy Search (MVE), Upper Confidence Bound (UCB), random sampling (rand.)).

improvement when initialized, indicating that while prior knowledge enhanced their performance, they remained functional even with limited initial data. In contrast, TS was largely unaffected by initialization, demonstrating robustness to starting conditions. Its randomized nature facilitated continuous exploration, allowing it to adapt over time even in the absence of a structured prior. Overall, these results highlight the varying dependence of acquisition functions on initialization: (i) EI and UCB strongly relied on an initial dataset, struggling without an informative prior. (ii) KG and MVE functions showed moderate sensitivity, benefiting from initialization but still effective without it. (iii) TS remained highly robust, performing well regardless of starting conditions.

Random sampling exhibited an initial improvement when initialized with k-means points; however, this advantage diminished rapidly after approximately 10 additional evaluations. While initialization provided a brief performance gain, random sampling remained consistently outperformed by all acquisition functions once further samples were collected. Notably, in the absence of initialization, both EI and UCB underperformed relative to random sampling during early iterations, further emphasizing their reliance on a structured prior.

**Performance of acquisition functions.** Fig 2 presents the comparative performance of acquisition functions in Bayesian optimization. For all experiments, the first 10 data points were selected using k-means clustering and were identical for all acquisition functions. This standardized initialization provides a consistent basis for comparing subsequent adaptive sampling strategies. Panel (A) illustrates the evolution of the NRMSE, $d_{x,y}$, and $d_\theta$ over the number of samples. Panel (B) summarizes the distribution of these metrics after 30 samples. Across all acquisition functions, the NRMSE, $d_{x,y}$, and $d_\theta$ decreased as more samples were acquired, albeit with varying rates of improvement: TS and KG achieved the lowest final NRMSE, consistently outperforming EI and UCB. MVE performed well initially, but resulted in higher final errors. Random sampling performed the worst, underscoring its inefficiency in guided optimization. Regarding spatial and angular accuracy, i.e., the distances $d_{x,y}$ and $d_\theta$ from the spatial and angular hotspot, TS and MVE demonstrated the best

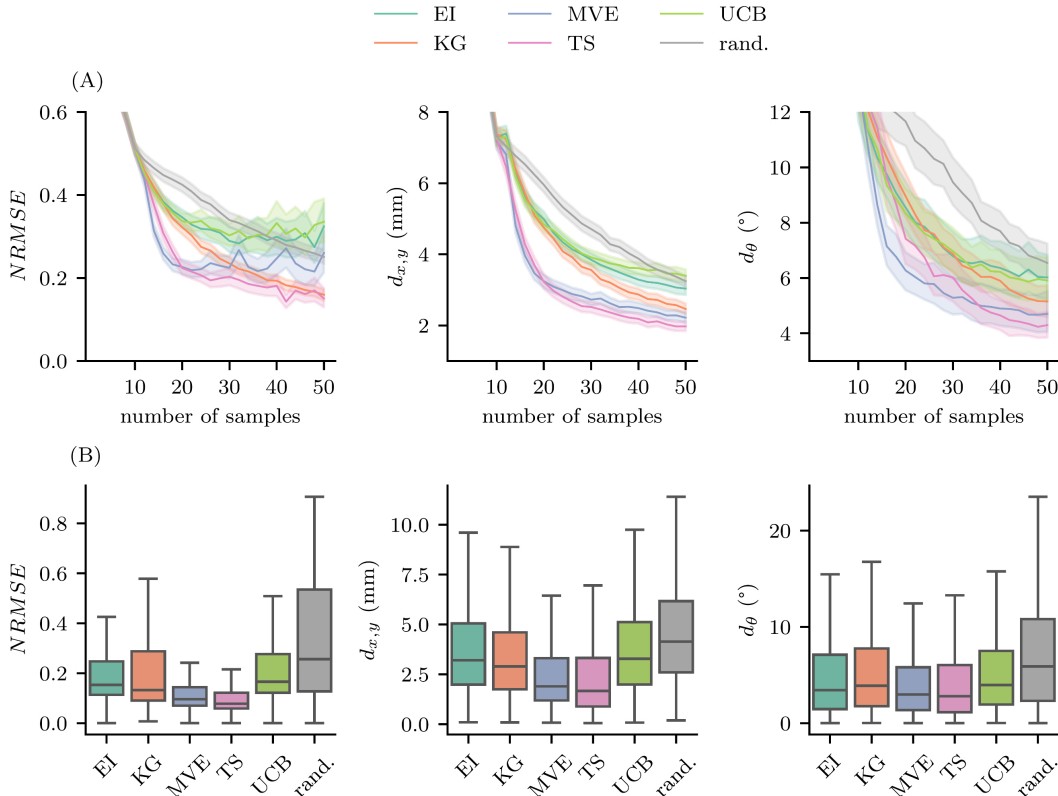

**Fig 2. Comparison of acquisition function performance.** For all eight participant, we subsampled from a total of 300 measured TMS trials and evaluate acquisition function performance using an initialization of 10 k-means grid points. (A) Normalized root mean square error NRMSE, positional error ($d_{x,y}$, and angular error ($d_\theta$) were tracked as the number of stimuli increased. Solid lines represent the mean, while the shaded area indicates the standard error of the mean across eight participants and 100 random seeds (800 runs in total). (B) After 30 samples, the distribution of NRMSE, $d_{x,y}$, and $d_\theta$ across all 800 runs is shown. Box plots indicate the interquartile range, with whiskers extending to the full data range (Expected Improvement (EI), Knowledge Gradient (KG), Max-Value Entropy Search (MVE), Thompson sampling (TS), Upper Confidence Bound (UCB), random sampling (rand.)).

performance, rapidly converging to the ground truth hotspot. UCB and EI exhibited the largest errors and the slowest convergence, offering only marginal improvement over random sampling. Fig 2(B) illustrates the performance of the acquisition functions after 30 samples with TS achieving the best overall performance, yielding the lowest median NRMSE, $d_{x,y}$, and $d_\theta$. After 30 stimuli, the median spatial distance to the hotspot was under 3 mm, and the median angular distance was below 5°. MVE performed similarly to TS, while UCB, EI, and KG lagged behind. Additionally, UCB, KG, and EI exhibited greater variance, suggesting increased sensitivity to initialization and individual differences. Importantly, across the considered experimental setup with identical k-means initialization, all acquisition functions outperformed random sampling.

**Acquisition functions exhibit distinct sampling strategies.** We analyzed the sampling strategies of the different acquisition functions, initializing each experiment with the same 10 k-means points. As shown in Fig 3, EI and UCB quickly converged to a local optimum after identifying a responsive area. However, this behavior was often suboptimal, as they failed to explore other potentially more responsive regions once convergence was reached. MVE followed a similar pattern, but it exhibited slightly more exploratory behavior. In contrast, KG was more exploratory, though its sampling distribution was skewed toward the edges of the input space, leading to an accumulation of null responses. TS, on the other hand, remained highly exploratory due to its stochastic sampling of posterior GP instantiations. Its samples remained

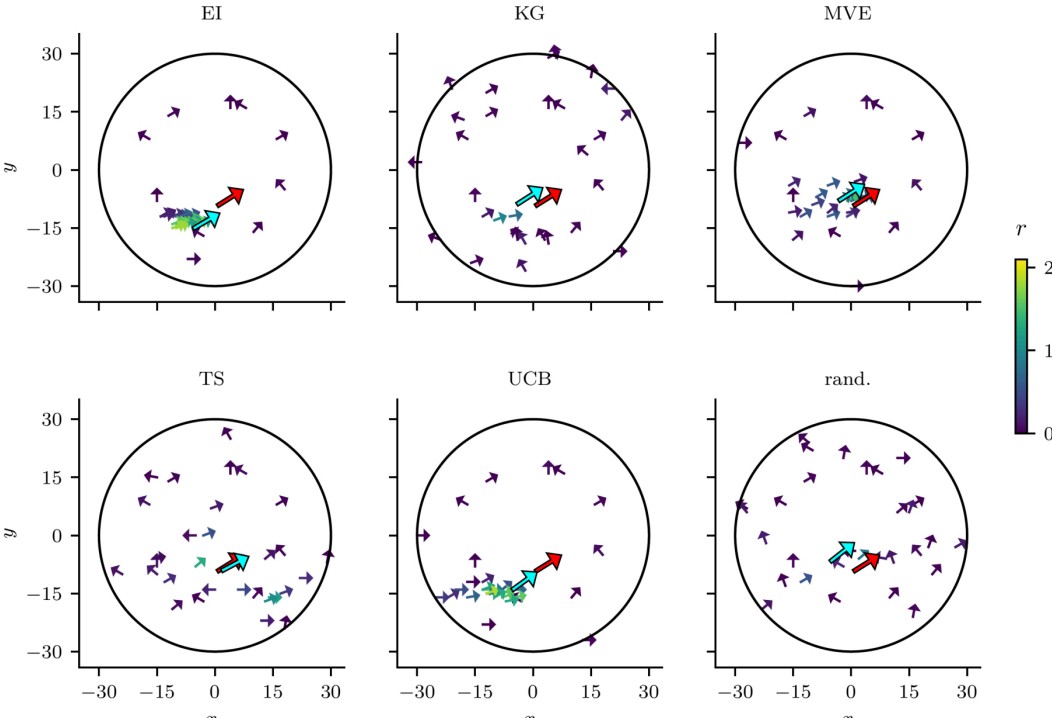

**Fig 3. Sampling strategies of acquisition functions.** Each subplot illustrates an exemplary sampling pattern of a specific acquisition function after 30 stimuli within the circular spatial constraint on stimulus locations. The $x$ and $y$ axes represent a normalized 2D coil placement coordinate system, which is projected onto the 3D scalp surface at the location of stimulation. Small arrows mark stimulus locations, with their orientation indicating the coil rotation angle. Arrow color encodes the response amplitude, as shown by the adjacent colorbar labeled 'r' (response amplitude, arbitrary units). The red arrow denotes the ground truth hotspot, while the blue arrow indicates the estimated hotspot identified by the acquisition function. Both the ground truth and estimated hotspots are defined in this 2D coordinate space, extended into 3D when including coil rotation. All subplots share the same 10 initialization points. Acquisition functions illustrated are Expected Improvement (EI), Knowledge Gradient (KG), Max-Value Entropy Search (MVE), Thompson Sampling (TS), Upper Confidence Bound (UCB), and random sampling (rand.).

centered around the target, exhibiting a favorable balance between exploration and exploitation. Notably, these results represent specific instances of the experiments and may vary due to the stochastic nature of Bayesian optimization with GPs.

## 4 Discussion

Localizing the motor hotspot is a crucial step in nearly all TMS applications, whether applied to healthy individuals or patient populations [22–34]. However, hotspot hunting remains a significant challenge, requiring the selection of an optimal coil configuration from a vast parameter space under strict time constraints. In this study, we introduced a novel approach that, to our knowledge, is the first to concurrently optimize both spatial and angular TMS parameters for motor hotspot hunting. Moreover, we systematically tested the performance of various acquisition functions for Bayesian optimization that, to date, have not been examined in the context of TMS-based motor hotspot hunting. A rigorous evaluation using electromyographic data consistently demonstrated that concurrent optimization is essential for improving the precision and efficacy of hotspot localization. By leveraging GPs and Bayesian optimization, our framework efficiently balanced exploration and exploitation across the extensive input space of stimulation parameters, overcoming limitations observed in previous studies [58,59,70–72].

Our findings demonstrate that fixing the coil rotation angle results in suboptimal hotspot hunting across all eight participants, reducing the expected maximum response by up to 29%. In contrast, treating coil rotation angle as an additional optimization parameter enabled a more precise localization of the motor hotspot. This approach facilitated a more comprehensive exploration of the input space, leading to improved estimates of the true motor cortical hotspot. Previous GP-based methods have either fixed key variables, such as maintaining a 45°coil rotation angle, or optimized spatial and angular parameters separately within a restricted range. These limitations reduce flexibility and fail to accommodate individual variability with the desired precision [70–72]. Our method addresses this issue by employing a comprehensive search strategy without imposing arbitrary constraints on coil rotation angle, thereby enhancing adaptability across individuals and experimental conditions. These results underscore the importance of simultaneously optimizing both spatial and angular TMS parameters in hotspot hunting.

Another key aspect of our study was the systematic evaluation of various acquisition functions for Bayesian optimization. This analysis provided mechanistic insights into the strengths and weaknesses of different parameter selection processes in the context of hotspot hunting. Acquisition functions play a crucial role in navigating the parameter space by balancing exploration and exploitation, determining which point the optimization algorithm should evaluate next based on the current predictive model. Our results suggest that acquisition functions such as EI and UCB tend to converge rapidly to local optima, as they primarily focus on exploitation. Consequently, they often fail to explore other potentially more responsive regions, making them less effective for TMS-based hotspot hunting, where premature convergence can lead to inadequate stimulation parameters. In contrast, TS and MVE achieved significantly better performance by adopting a more exploratory approach, thereby maintaining a more optimal balance between exploration and exploitation. Notably, TS consistently produced the best outcomes, identifying hotspots with the highest spatial precision and demonstrating robustness to initialization variability. Its efficiency in navigating large search spaces makes it a strong candidate for practical TMS applications. These findings highlight the importance of employing flexible, data-driven sampling methods to enable a comprehensive exploration of the input space and obtain more accurate estimates of the true motor cortical hotspot.

Enhancing the spatial and angular precision of TMS-based hotspot hunting has wide-ranging implications for neuroscience and clinical practice. More than 90 % of rTMS studies determine stimulation intensity based on motor cortex stimulation, even when targeting other regions such as the dorsolateral prefrontal cortex [83]. By improving hotspot localization, we provide a more precise and reproducible basis for estimating motor hotspots and, indirectly, motor thresholds, thereby enabling more accurate dosing in both research and clinical applications. In this way, advances in motor hotspot hunting extend well beyond motor cortex mapping, supporting the reliability and effectiveness of neuromodulatory interventions more broadly.

Our approach is particularly well-suited for neuronavigated, robotic-arm-assisted closed-loop TMS applications, which are increasingly employed in neuroscience research and clinical treatments for psychiatric disorders such as medication-resistant depression and neuropediatric conditions [59,69,84–100]. Importantly, our approach is not restricted to the motor cortex; it is broadly applicable to other TMS applications requiring parameter optimization, facilitating hotspot identification beyond the motor cortex. For instance, combining this approach with alternative readouts such as electroencephalography could facilitate hotspot identification not only within the motor cortex but also beyond it, extending its utility across a wide range of neuroscientific and clinical studies [101–103].

A key limitation of our approach is the absence of a CE-certified commercial neuronavigation system capable of defining the next target during neuronavigation (for an overview, see [104]). Consequently, our method is currently limited to non-CE-certified interim solutions. While our approach holds promise for clinical applications, its integration into routine practice remains challenging, primarily because it demands full automatization and CE-certification. Future studies should explore optimizing both the motor hotspot and motor threshold using GPs and Bayesian optimization, to further streamline the clinical application. Investigating alternative acquisition functions and refining closed-loop optimization strategies may further enhance the robustness and adaptability of hotspot hunting for diverse TMS applications. The method proposed in

 

the present work could be integrated into already existing toolboxes such Brain Electrophysiological recording and STimulation (BEST), building on their already powerful capabilities and further enhancing them by providing improved hotspot identification and parameter optimization [105]. Our results indicate that approximately 50 stimulation points are sufficient to achieve accurate hotspot localization, aligning well with practical clinical and robotic neuronavigation settings.

In conclusion, our findings demonstrate that concurrently optimizing spatial and angular TMS parameters is essential for efficient and spatially precise hotspot hunting. Based on our results, we recommend using acquisition functions that prioritize exploration—particularly TS—to prevent premature convergence and ensure robust hotspot identification. This study represents a significant advancement toward a more comprehensive understanding of TMS cortical targets, with important implications for clinical practice and research applications, including motor mapping, pre-surgical planning, and personalized neuromodulation therapies.

## Acknowledgments

The authors thank Jasper Hoffmann for proofreading the manuscript and providing valuable feedback. The authors thank Linus Fritz for conducting initial experiments as part of his bachelor's thesis. The academic English of this paper was improved with the assistance of ChatGPT (version 4.1). The experimental design, content, analyses, and interpretations presented in the paper were entirely the work of the authors and not influenced by ChatGPT.

## Author contributions

**Conceptualization:** David Luis Schultheiss.

**Data curation:** David Luis Schultheiss, Zsolt Turi.

**Formal analysis:** David Luis Schultheiss.

**Funding acquisition:** Joschka Boedecker, Andreas Vlachos.

**Investigation:** David Luis Schultheiss, Zsolt Turi.

**Methodology:** David Luis Schultheiss.

**Project administration:** David Luis Schultheiss, Zsolt Turi.

**Resources:** Joschka Boedecker, Andreas Vlachos.

**Software:** David Luis Schultheiss.

**Supervision:** Joschka Boedecker, Andreas Vlachos.

**Validation:** David Luis Schultheiss, Zsolt Turi.

**Visualization:** David Luis Schultheiss.

**Writing – original draft:** David Luis Schultheiss, Zsolt Turi.

**Writing – review & editing:** David Luis Schultheiss, Zsolt Turi, Joschka Boedecker, Andreas Vlachos.

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
