## [Decision Letter · Decision Letter 0]

3 Sep 2025

PCOMPBIOL-D-25-00770

Efficient Gaussian Process-based Motor Hotspot Hunting with Concurrent Optimization of TMS Coil Location and Orientation

PLOS Computational Biology

Dear Dr. Schultheiss,

Thank you for submitting your manuscript to PLOS Computational Biology. After careful consideration, we feel that it has merit but does not fully meet PLOS Computational Biology's publication criteria as it currently stands. Therefore, we invite you to submit a revised version of the manuscript that addresses the points raised during the review process.

Please submit your revised manuscript within 60 days Nov 03 2025 11:59PM. If you will need more time than this to complete your revisions, please reply to this message or contact the journal office at ploscompbiol@plos.org. Please include the following items when submitting your revised manuscript:

We look forward to receiving your revised manuscript.

Kind regards,

Ming Bo Cai

Academic Editor

PLOS Computational Biology

Joseph Ayers

Section Editor

PLOS Computational Biology

**Journal Requirements:**

At this stage, the following Authors/Authors require contributions: David Luis Schultheiss, Zsolt Turi, Joschka Boedecker, and Andreas Vlachos. Please ensure that the full contributions of each author are acknowledged in the "Add/Edit/Remove Authors" section of our submission form.

4) Please ensure that all Figure files have corresponding citations and legends within the manuscript. Currently, Figure 4 in your submission file inventory does not have an in-text citation. Please include the in-text citation of the figure.

Potential Copyright Issues:

i) Figure 4. Please confirm whether you drew the images / clip-art within the figure panels by hand. If you did not draw the images, please provide (a) a link to the source of the images or icons and their license / terms of use; or (b) written permission from the copyright holder to publish the images or icons under our CC BY 4.0 license. Alternatively, you may replace the images with open source alternatives. See these open source resources you may use to replace images / clip-art:

6) Your current Financial Disclosure states, "The author(s) received no specific funding for this work."

However, your funding information on the submission form indicates receiving a fund. Please ensure that the funders and grant numbers match between the Financial Disclosure field and the Funding Information tab in your submission form. Note that the funders must be provided in the same order in both places as well.

Please amend your detailed Financial Disclosure statement. This is published with the article. It must therefore be completed in full sentences and contain the exact wording you wish to be published.

1) Please clarify all sources of financial support for your study. List the grants, grant numbers, and organizations that funded your study, including funding received from your institution. Please note that suppliers of material support, including research materials, should be recognized in the Acknowledgements section rather than in the Financial Disclosure

2) State the initials, alongside each funding source, of each author to receive each grant. For example: "This work was supported by the National Institutes of Health (####### to AM; ###### to CJ) and the National Science Foundation (###### to AM)."

3) State what role the funders took in the study. If the funders had no role in your study, please state: "The funders had no role in study design, data collection and analysis, decision to publish, or preparation of the manuscript."

4) If any authors received a salary from any of your funders, please state which authors and which funders.

7) Thank you for stating "The authors declare no conflict of interest." Please revise your current Competing Interest statement to the standard "The authors have declared that no competing interests exist."

**Reviewers' comments:**

Reviewer's Responses to Questions

Reviewer #1: “A systematic review found that over 90% of studies determine rTMS intensity based on motor cortex stimulation, even when the intended cortical target is elsewhere (e.g., the dorsolateral prefrontal cortex) [83]. Our method has the potential to improve rTMS target selection by enhancing spatial precision and reproducibility, ultimately leading to more effective neuromodulatory interventions.” Here are two argments likely incorrect connected. A more efficient hotspot identification does not solve the problem of using it as a target elsewhere.

Accordingly here “Importantly, the proposed method is not restricted to the motor cortex; it is broadly applicable to other TMS applications requiring parameter optimization, facilitating hotspot identification beyond the motor cortex.” It should be clearly stated that other outreads such as TMS-EEG are needed.

“is the absence of a CE-certified commercial neuronavigation system capable of defining the next target during neuronavigation” agreed. However two amendments might be helpful here. 1. An open source availability of the present programm might be helpful (or not??) just by advising “move coil xx cm towards…. and angulate it by xx ° clock- or counterclockwise…. Would the time consumption by manually feeding this program be worth the effort?. 2. Are there any advices for todays researchers without the program simply using conventional hotspot hunting for the moment?

Reviewer #2: The authors propose a Gaussian process Bayesian‐optimization framework that jointly optimizes both TMS coil position and orientation, and compare five acquisition functions on eight subjects’ EMG data and on simulated data. They conclude that Thompson sampling is most robust. The idea is clear, the implementation should be open source, and the work has methodological and practical value. However, there are still some problems in the analysis in this paper. I recommend major revisions to strengthen validation, clarify methods, and temper claims before acceptance.

Individual points:

1. There is a redundant) in line 96.

2. The results of random sampling is presented in Figure 1. However, in the corresponding analysis of Figure 1 (in text), the random sampling is not mentioned or compared with the other methods.

3. The results in Figure 2 A do not present the data for the first 10 steps. The distribution of the starting point should have an influence on the performance of these methods and should not be omitted in the analysis.

4. In the analysis of Figure 2B, you say

“All acquisition functions outperform random sampling.”

But this is inconsistent with the performance of random sampling in Figure 1.

5. The legend of Figure 3 is not obvious enough.

6. Please clearly states your advantage compared to the previous Bayesian neuronavigation papers. For example:

https://doi.org/10.1016/j.neuroimage.2015.09.013

https://doi.org/10.1016/j.neuroimage.2020.117082

https://doi.org/10.1016/j.neuroimage.2017.04.001

That should be a quantitative comparison that also looks into the mechanisms so that readers could really learn something and not just see yet another Bayesian method for the same without further information why and which one for what.

7. The abstract mentions “generative models” for data augmentation, but the manuscript lacks specifics on model type, parameters, or validation. Describe the simulation pipeline in detail and address potential biases. Moreover, if possible, can you describe the parameters of your 5 acquisition functions?

8. There are previous models for synthetic TMS motor responses with population statistics out there. A comparison would be great. A test with several of some such models would be even better as they might also increase the n, which could be a bit higher.

9. The simulations use 300 stimuli and evaluate ~2800 test points—this is maybe unrealistic in clinical settings. Is it possible to discuss performance under <100 stimuli and the time/cost trade off in conjunction with robotic navigation.

**Have the authors made all data and (if applicable) computational code underlying the findings in their manuscript fully available?**

Reviewer #1: None

Reviewer #2: Yes

PLOS authors have the option to publish the peer review history of their article (what does this mean?). If published, this will include your full peer review and any attached files.

Reviewer #1: No

Reviewer #2: No

**Figure resubmission:**
---

## [Decision Letter · Decision Letter 1]

6 Feb 2026

Dear Schultheiss,

We are pleased to inform you that your manuscript 'Efficient Gaussian Process-based Motor Hotspot Hunting with Concurrent Optimization of TMS Coil Location and Orientation' has been provisionally accepted for publication in PLOS Computational Biology.

Best regards,

Ming Bo Cai

Academic Editor

PLOS Computational Biology

Joseph Ayers

Section Editor

PLOS Computational Biology

Thanks for addressing the reviewer comments. We apologize for the slow response. Although not all previous reviewers are available to review the revision, the editor reads through the responses and judges that the concerns have been sufficiently addressed. As pointed out by the authors during this process, a related preprint appeared on bioRxiv. You are welcome to include comments about that work in your final manuscript if you desire.

Reviewer's Responses to Questions

**Comments to the Authors:**

Reviewer #1: none

**Have the authors made all data and (if applicable) computational code underlying the findings in their manuscript fully available?**

Reviewer #1: None

PLOS authors have the option to publish the peer review history of their article (what does this mean?). If published, this will include your full peer review and any attached files.

Reviewer #1: No

---

## [Editor Report · Acceptance letter]

PCOMPBIOL-D-25-00770R1

Efficient Gaussian Process-based Motor Hotspot Hunting with Concurrent Optimization of TMS Coil Location and Orientation

Dear Dr Schultheiss,

I am pleased to inform you that your manuscript has been formally accepted for publication in PLOS Computational Biology. Your manuscript is now with our production department and you will be notified of the publication date in due course.

With kind regards,

Judit Kozma
